# Infective Endocarditis—Characteristics and Prognosis According to the Affected Valves

**DOI:** 10.3390/microorganisms12050987

**Published:** 2024-05-14

**Authors:** Bistra Dobreva-Yatseva, Fedya Nikolov, Ralitsa Raycheva, Mariya Tokmakova

**Affiliations:** 1First Department of Internal Medicine, Section of Cardiology, Cardiology Clinic, Faculty of Medicine, Medical University—Plovdiv, UMBAL “St. Georgi” EAD, 4002 Plovdiv, Bulgaria; fedyani@yahoo.com (F.N.); m14021971@yahoo.co.uk (M.T.); 2Department of Social Medicine and Public Health, Faculty of Public Health, Medical University—Plovdiv, 4002 Plovdiv, Bulgaria; dirdriem@gmail.com

**Keywords:** infective endocarditis, affected valve, mortality

## Abstract

Background: Infective endocarditis (IE) continues to be a disease with high mortality despite medical advances. Objective: The objective of this study was to investigate the characteristics and prognosis of IE according to the affected valves. Materials and methods: This study was retrospective and single-centered, and it included 270 patients with a diagnosis of IE, for the period 2005–2021, who received treatment at the University Hospital “St. Georgi” in Plovdiv, Bulgaria. Results: Single-valve IE (SIE) was found in 82.6% (n-223), multivalvular IE (MIE) in 16.66% (n = 45) and device IE (CDRIE) in 0.74% (n = 2) of patients. The most commonly affected valve was the aortic valve, in 44.8% (n = 121). The predominant multivalvular involvement was aortic–mitral valves (AV-MV) (13.7%, n = 37). The patients with tricuspid valve (TV) IE were significantly younger, at 39 (30) years, and were more frequently male (80.8%). Mortality was higher in MIE than in SIE (31.1% vs. 23.8%) and was the highest in multivalve aortic–tricuspid (AV-TV) IE (75%). Early surgery was performed most in AV-MV IE, in 29.7% (n = 11). The Charlson comorbidity index (CCI) was significantly higher in MV 4 (4) and AV 3 (3) vs. TV IE 1 (5) (*p* = 0.048 and *p* = 0.011, respectively). Septic shock occurred most frequently in AV-TV involvement (75%; *p* = 0.0001). The most common causative agents were of the *Staphylococcus* group. *Staphylococcus aureus* more often affected TV alone (46.2%, n = 124) vs. AV (9.9%, n = 14; *p* = 0.0001) and vs. MV (22.6%, n = 17; *p* = 0.022); *Staphylococcus coagulase-negative (CNG)* was the prevalent cause of MV IE (22.7%, n = 17) vs. AV-MV (2.7%, n = 1; *p* = 0.007). *Streptococci* were represented in a low percentage and only in left-sided IE, more frequently in AV-MV (18.9%, n = 7) vs. AV (6.6%, n = 8; *p* = 0.025). Conclusions: The aortic valve is the most frequently affected valve, as single-valve IE or as multivalve AV-MV, with the predominant causative agents being of the *Staphylococcus* group. AV-TV IE has the worst prognosis, with the most common complication of septic shock and the highest in-hospital mortality.

## 1. Introduction

IE is a disease with poor prognosis and has had a changing profile over the past few decades. The incidence of IE has increased from 9.91 to 13.8/100,000 in the last 30 years [1]. Despite advances in diagnosis, including new imaging modalities, and in treatment with the use of new antibiotic molecules and early surgical treatment, mortality in IE remains high at 16–25% [2,3]. The characteristics of IE change over time and depend on the geographical and socio-economic level of the country. In recent decades, there has been an increase in the patients’ age, comorbidity, *Staphylococcus* etiology, and health-care-related IE. There is a shift in predisposing factors with an increase in cases of degenerative valvular involvement, valve protheses, indwelling catheters, and implanted cardiac devices, while cases due to rheumatic heart disease are becoming less common in developed countries. Although IE is more common in older people, the disease can affect younger patients, usually those reporting intravenous drug use (IDU) or congenital heart disease [2,3]. Patients with IDU-IE have a demographic and clinical profile different from that of non-IDU-IE. They are younger, with fewer comorbidities, and hepatitis C virus (HCV) infection is frequent (36–82%) [4]. The most commonly involved valve is the tricuspid valve (58–77%), with predominant *Staphylococcus aureus* infection [5,6,7].

IE usually affects one valve and mostly on the left side, less commonly two valves (10–18%), and extremely rarely three or four valves [3,8,9]. There are only 16 reported cases of triple-valve IE [10]. Left-sided IE has a poor prognosis; it is more often complicated by heart failure and embolic events, and more frequently there are indications for early surgical treatment. According to the Euro-Endo Registry, only half of the indicated patients are operated on, and this behavior improves the prognosis [11]. Right-sided IE accounts for 5–10% and is more often associated with intravenous drug use, intracardiac devices, and central venous catheters [12]. Some studies have compared single- and multivalve IE regarding the outcome and prognosis [8,13,14,15]. Kim et al. found 18% MIE, with involvement of the mitral and aortic valves being most common. The mortality rate was similar between SIE and MIE, but heart failure complications were significantly more frequent in MIE than SIE [13]. Using data from the Spanish IE Registry, Alvarez-Zaballos et al. reported that MIE is associated with higher incidence rates of intracardiac complications, heart failure, surgical indications, surgery, and in-hospital mortality than SIE. MIE, but not SIE, was the independent predictor for in-hospital mortality in this group [8]. Bohbot et al. compared single- to multivalve left-sided IE and reported that patients with bivalvular IE had more embolic events, congestive heart failure, vegetations, and perivalvular complications than did those with monovalvular IE. Early surgery was more frequent for patients with bivalvular IE. The thirty-day mortality rate was higher for patients with bivalvular IE than for those with monovalvular IE. Early surgery was associated with increased survival for patients with bivalvular IE [9].

Data on the characteristics and outcomes of IE according to the affected valves are scarce in our region. This information could serve to improve the prognosis and outcomes in IE, identify high-risk patients, and enable more accurate initial antibiotic therapy and decisions for more aggressive treatment with early surgical intervention.

## 2. Materials and Methods

The study was retrospective and single-centered, including 270 patients with a diagnosis of IE according to the modified Duke criteria, treated at the University Hospital “St. Georgi” in the city of Plovdiv during the period of January 2005–December 2021. The hospital capacity is 1500 beds, and the cardiology clinic is a reference center for the treatment of IE for a large part of southern Bulgaria. The medical records of treated patients with codes I33, I38, and I39 for the described period were used. The variables studied included the patients’ demographics, risk group, presence of predisposing heart disease, comorbidities, Charlson comorbidity index (CCI) [16], entry point, predictors for transient bacteremia, clinical and echocardiographic findings, causative organisms, complications, and clinical outcomes.

### 2.1. Definition and Classification of IE

The diagnosis was defined as definite IE or possible IE according to the modified Duke criteria [6]. Surgical treatment of IE was defined as early when the surgery was performed during antibiotic treatment. Valvular involvement of IE was determined based on findings from echocardiography, other imaging studies, cardiac surgery, or, in some cases, clinical presentation. The presence of septic emboli and an extracardiac focus of infection was defined as a focus of infection detected by imaging or based on a typical clinical presentation. Complications were diagnosed according to the established diagnostic criteria and recommendations.

### 2.2. Statistical Methods

Quantitative data are presented as the arithmetic mean ± standard deviation (mean ± SD) or median and interquartile range (median (IQR)) according to the type of distribution of the variables (Kolmogorov–Smirnov test). Categorical variables were summarized using absolute (n) and relative (%) magnitudes. The Mann–Whitney test for independent samples was used to compare quantitative variables between two groups. A z-test was used to compare the relative shares of categorical variables between the studied groups. A *p*-value of <0.05 (two-tailed test) was considered statistically significant for all tests. Statistical analysis was performed using SPSS, version 26.0 (IBM Corp., New York, NY, USA).

## 3. Results

Of all 270 patients, 75.9% (n = 205) had definite IE, with 65% (n = 133) of them having two major criteria and 35% (n = 72) having one major and three minor criteria. The remaining 24.1% (n = 65) were diagnosed with possible IE, 95% (n = 62) of them with one major and one minor criterion and three with three minor criteria.

Transthoracic echocardiography was performed in 100% of patients and transesophageal echocardiography in 35.9% of them. We found vegetation in 83.7% (n = 226), periannular abscess in 3% (n = 8), chordal rupture in 3.3% (n = 5), and valve obstruction in 11.9% (n = 32). The distribution of valvular regurgitation according to severity was as follows: mild/moderate: AV—29.3%, MV—23.7%, TV—6%; severe regurgitation: AV—24.8%, MV—19.3%, TV—7.4%.

Single-valve IE (SIE) was found in 82.6% (n = 223), multivalvular IE (MIE) in 16.66% (n = 45), and CDRIE in 0.74% (n = 2). The most frequently affected valve was the aortic (AV—44.8%, n = 121), followed by the mitral valve (MV—27.8%, n = 75) and tricuspid valve (TK—9.62%, n = 26). We had one case (0.37%) with pulmonary valve IE. Of the MIE types, the most common was AV-MV IE (13.7%, n = 37) (Table 1). There was no triple- or quadruple-valve endocarditis in our series.

The TV IE patients were younger, at 39 (30) years (statistically significant differences are presented in Table 1). The male gender predominated in all valvular involvements, with the highest proportion in TV IE (80.8%) and AV-TV IE (100%) (*p* = 0.019) (Table 1).

The thirty-day mortality was higher in MIE than in SIE (31.1% vs. 23.8%), without statistical difference (*p* > 0.05). Mortality was significantly higher in AV-TV IE (75%), in PV IE (100%), and in CDRIE (100%) (statistically significant differences are presented in Table 1).

The Charlson comorbidity index (CCI) was the highest in MV IE 4 (4) and AV-MV IE 4 (3) and the lowest in TV IE—1 (5) (AV vs. TV (*p* = 0.048); MV vs. TV (*p* = 0.011)). Early surgery was most often performed for CDRIE (100%) and multivalve AV-MV IE (29.7%) and was statistically significantly more frequent compared to MV IE (13.3%) (*p* = 0.036). Of the SIE types, the most frequent early surgery was performed in TV IE (23.1%).

Native IE predominated in all valvular involvements, being more common in TV (80.8%), PV (100%), and AV-TV (100%). Prosthetic IE accounted for nearly one-third of all cases, most commonly in left-side IE: AV (38.8%), MV (25.3%), and AV-MV (35.1%). Valvular prothesis was the most common cardiac predisposing condition, with no significant differences between types of valvular involvement. Degenerative valvular lesions were most often associated with AV IE (8.3%). Bicuspid AV occurred in 7.4% and MV prolapse in 6.7%. In about half of cases, there was no confirmed predisposing heart disease.

The portal of entry of infection was unknown in about 50% of cases. In TV IE, the most common portal of entry was IV drug use (61.5%). The next known gateway was manipulations and procedures in all cases of SIE, in AV-MV IE, and in device IE. The dental portal was a source of infection only in left-sided IE, most often in AV-MV (18.9%), followed by AV (12.4%) and MV (10.7%). Hemodialysis was a common cause of MIE (AV-TV 25%; MV-TV 25%), as well as single-valve involvement of AV and MV.

Community-acquired IE was most common in left-side IE—AV (71.9%), MV (69.3%), and AV-MV (70.3%). Health-care-associated IE involved all valves, significantly more frequently in CDRIE (100%) and MV-TV IE (50%) (*p* < 0.05). Intravenous-drug-related IE most often affected TV (57.7%) and AV-TV (50%) (*p* = 0.0001) (Table 1).

The Charlson comorbidity index (CCI) was significantly higher in MV IE 4 (4) and AV IE 3 (3) vs. TV IE 1 (5) (*p* = 0.048 and *p* = 0.011, respectively) and in AV-MV 3.5 (3). Arterial hypertension was the most common comorbidity in left-side IE—AV (69.4%), MV (69.3%), and AV-MV IE (56.8%) vs. TV IE (30.8%) (*p* = 0.0002, *p* = 0.001, and *p* = 0.041, respectively). Coronary heart diseases were prevalent in AV IE (30.6%) vs. TV (7.7%) (*p* = 0.016). Hemodialysis was frequent in MIE—AV-TV (25%) and MV-TV (25%). Chronic liver diseases most often accompanied TV IE (19.2%) vs. AV (5.8%) and vs. MV (1.3%) (*p* = 0.024 and *p* = 0.001, respectively). A past IE was most common in TV (26.9%) vs. AV (4.1%) and vs. MV (4.0%) (*p* = 0.0001 and *p* = 0.0008, respectively). There were no significant statistical differences in the remaining comorbidities among the various valvular involvements (Table 2).

The most common complication was acute heart failure, followed by impaired renal function, embolic events, and acute stroke, with no significant differences among the valve locations. Septic shock was significantly more frequent in AV-TV involvement (75%) vs. AV (9.1%) and vs. MV (6.7%) (*p* < 0.0001 and *p* < 0.0001, respectively) and also when compared to both TV (7.7%) (*p* = 0.001) and AV-MV (5.4%) (*p* < 0.0002) (Table 2).

Negative blood cultures were prevalent in AV IE (50.4%) compared to TV IE (23.1%) (*p* = 0.011). The *Staphylococcus* group was the predominant cause of IE (TV 53.8%, MV 45.3%, and AV-TV 50%). *Staphylococcus aureus* significantly more often affected TV (46.2%) compared to AV (9.9%) (*p* = 0.0001) and compared to MV (22.6%) (*p* = 0.022); *Staphylococcus CNG* was the most considerable part of the etiology of MV IE (22.7%) compared to AV-MV (2.7%) (*p* = 0.007) and that of CDRIE (50%) compared to AV-MV (2.7%) (*p* = 0.003). *Streptococci* were represented in a low percentage and were the causative agent only of left-sided IE, being significantly more frequent in AV-MV IE (18.9%) compared to AV (6.6%) (*p* = 0.025). *Enterococci* were found in all valvular involvements but most often in MIE, in MV-TV (50%), AV-TV (25%), PV (100%), and CDRIE (50%), compared to SIE (vs. AV 7.4%, *p* = 0.003; vs. MV 6.7%, *p* = 0.003; vs. TV 7.7%, *p* = 0.021). *Enterococcus faecalis* was the most widespread agent of this group. Gram-negative (non-HASEK) bacteria were the most frequent causative agent of single-valve involvement, with *Escherichia coli* being the most common representative (14%). *Klebsiella pneumoniae* affected only AV (1.7%) and MV (1.3%) (Table 3).

## 4. Discussion

Our data present a prevalence of single-valve IE, with predominant involvement of the aortic valve. The results from EURO-ENDO (49.5%) [3], Latin America (42.4%) [17], and Canada (40%) [18] are similar. The highest frequencies of infection of the MV were reported in ICI-PCS (41%) [2], Africa (54.7%) [19], South Korea (61.3%) [20], Japan (42.2%) [21], Turkey (43.3%) [22], and Vietnam (41.3%) [23]. The distribution of affected valves is associated with the heart valve predisposition, entry point, and causative microorganism. Predisposing cardiac conditions are an important part of IE pathogenesis. Their spectrum and distribution have undergone a substantial change over the last few decades, with significant differences in the geographical and socio-economic status of the countries also observed. Rheumatic heart disease (RHD) was the most common underlying lesion in the past, and the mitral valve was the most affected, with a *Streptococcus* etiology [24]. In developed countries, the proportion of cases associated with RHD has declined to 5% or less over the past two decades [3]. Nowadays, the most affected valve is the aortic valve as a result of degenerative valve disorders [25]. However, in developing countries, RHD remains the most common predisposing risk factor for IE. For example, a meta-analysis for Africa published in 2022 reported 52% of cases with a predisposition of RHD [19], India reported 19% [26], Saudi Arabia reported 15% [27], and Latin America reported 13% [17]. For comparison, in the International Collaboration on Endocarditis Prospective Cohort Study (ICE-PCS), cases with RHD were registered at 3% [2]. We found that patients with AV IE were older, with more frequent degenerative valve predisposition, a high CCI, a predominantly Staphylococcus group etiology, and prevalent comorbidities of arterial hypertension and coronary heart disease.

The mitral valve was the second most frequently affected valve, characterized by older patients with the highest CCI, predominantly community-acquired IE, and *Staphylococcus CNG* as the causative agent.

We found tricuspid valve involvement in 9.6%, which is comparable to the data from other studies: EURO-ENDO—11.4% [3], ICI-PCS—12% [2], Latin America—9.3% [17], and Africa—7.2% [19]. These patients were significantly younger, predominantly male, with a low CCI [16] and more frequent involvement of a native valve; additionally, 61.2% were IV drug users. Chronic liver diseases were most common in TV IE, due to IV drug users with hepatitis C virus infection, which is frequent (36–82%) among these patients [28]. The predominant causative microorganism was *Staphylococcus aureus*. In this period, we had only two cases of CDRIE, with VVI pacemakers. Pulmonary valve IE is a rare entity, comprising 1–2% of all cases, which is in accordance with our results [29].

Multivalve IEs were found in 16.7% of our patients, which is similar to the findings reported by EURO-ENDO (18.2%) [2], Iran (17%) [30], South Korea (16.7%) [20], Latin America (13.2%) [17], and India (13.2%) [26]. Controversially, other countries’ data include fewer cases: Japan—6.1% [21], Turkey—7.7% [22], Vietnam—9.5% [23]. This difference may be due to the different number of cases, level of diagnostic techniques, frequency of use of TEE, and other new diagnostic modalities. In our study, TEE was performed in 35.9% of patients. The outcomes from Canada are comparable—29.4% [8]. TEE was used more often in Japan (73.3%) [21], Latin America (59.6%) [17], ICE-PCS (59%) [2], EURO-ENDO (58.1%) [3], and Iran (54.4%) [30]. TEE was performed significantly less frequently in India—18.1% [26]. Insufficient data exist regarding the distribution of MIE. We found the involvement of AV-MV to be the most frequent, which is similar to other reports [8,9,13].

The age of patients with IE has increased over the past decades, and our results are like those from the EURO-ENDO registry: a mean age of 59.25 ± 18.03 years (46.3% aged >65 years and 12.0% aged ≥80 years) and higher in European than in non-European countries (60.97 ± 17.36 vs. 52.66 ± 19.01, *p* < 0.0001) [3]. The data from other economically developed countries currently exhibit similarities: France—69 years [31], Japan—69.1 years [21], Canada—56 years [18], Spain—61.8 years [32], Portugal—47.1% between 60 and 79 years old [33], Netherlands—67.5 years [20,34], and South Korea—56 years [20]. Only the patients with tricuspid valve IE were significantly younger, relating to drug abuse. The male gender remains the predominant gender. This has not changed over the years.

The entry door of infection is related to the affected valves, causative microorganisms, and type of acquisition. In half of all cases, the portal of entry is unknown. The portal of entry was dental only in left-side IE, with low frequency in AV, MV, and AV-MV, most often with a *Streptococcal* etiology and community-acquired IE. The *Streptococcal* etiology has decreased over the past decades, and this study produced results that corroborate the findings of a great deal of the previous work in this field in the EURO-ECHO registry. The most common causative agents were of the *Staphylococcus* group. *Staphylococcus aureus* more often affected TV alone, while *Staphylococcus coagulase-negative (CNG)* was the prevalent cause of MV IE. Manipulations and procedures are presented as a portal of entry in all types of valve involvement and are a predisposing factor for health-care-associated IE. This type of IE presented with the most common staphylococcal etiology and an increasing proportion of Enterococcus bacteria as a causative agent [15].

Prosthetic valves were the most prevalent cardiac predisposition, in nearly one-third of all cases, without significant differences according to the valve location. These results are consistent with those of European and other economically developed countries and suggest an increase in prosthetic valve IE (PVIE) cases. In comparison, cases with PVIE accounted for 21% in ICE-PCS [2], 25% in France [31], 26% in Euro Heart Survey [35], and 30% in EURO-ENDO [3].

The CCI is higher in the left-side IE, due to advanced age, more common arterial hypertension and coronary heart disease, and extensive contact with the health care system. Hemodialysis is a predominant comorbidity and predisposition in MIE, particularly AV-TV and MV-TV, with mostly Staphylococcus and Enterococcus etiologies, because of health-care-acquired IE [15].

We found higher 30-day mortality rates in multivalve IE versus single-valve IE, without a statistical difference. The prognosis of MIE is unclear. A poor outcome has been described in some studies [15,36,37,38] but not in others [13,39,40,41]. We found a significantly higher mortality rate in AV-TV and CDRIE than in the other valve involvements, despite the small number of patients in these groups. The most frequent complications were heart failure, embolic events, impaired kidney function, and stroke, similar to the other studies. We did not identify any difference in complication rates according to valve involvement, except for septic shock. The significantly higher rate of septic shock in AV-TV correlates with the higher mortality in this group. Septic shock is associated with high mortality in IE [42,43]. We did not register any differences in 30-day mortality regarding the different types of MIE.

## 5. Limitation

The study was retrospective and the data were based on the clinical database of a single center. We did not use nuclear imaging diagnostics (18F-fluorodeoxyglucose positron emission tomography or leucocyte scintigraphy) due to the unavailability of these resources. This is a very important study, and major criteria for diagnosis, especially in cases of prosthetic IE, and minor criteria for cases of extracardiac foci of infection were found. In cases of negative blood culture, these data are crucial for diagnosis. Another limitation of our study is that only in-hospital follow-up was available, and we do not know what impact the different valve involvements had on late survival. Despite these limitations, our study is the only such study performed in Bulgaria in the last several decades, and it included a large number of patients over a long period of time (17 years).

## 6. Conclusions

In patients with IE in our region, the most affected valve is the aortic valve, either as a single- or multivalve location; such patients also tend to be in older, with a high CCI, and with mostly *Staphylococcus* or *Enterococcus* etiology. Multivalve aorto-tricuspid IE has the worst prognosis, with a significantly high rate of septic shock and the highest in-hospital mortality. Knowing the specificities of IE according to the affected valve would help in the more accurate selection of the initial empiric antibiotic treatment. More aggressive therapeutic behavior and the decision for early surgical treatment in cases with poor prognostic features would improve the prognosis for these patients.

## Figures and Tables

**Table 1 microorganisms-12-00987-t001:** Clinical characteristics—demographics, risk group, outcome at 30 days (death), EF, type of valves, early surgery, entry point, predisposing heart conditions, type of acquisition.

Variables	AV	MV	TV	PV	AV + MV	AV+ TV	MV + TV	CDRIE	*p*-Value
	n = 121 (44.81%)	n = 75 (27.80%)	n = 26 (9.62%)	n = 1 (0.37%)	n = 37 (13.70%)	n = 4 (1.48%)	n = 4 (1.48%)	n = 2 (0.74%)	
	1	2	3	4	5	6	7	8	
	Single-Valve IE—223 (82.60%)				Double-Valve IE— 45 (16.66%)			2 (0.74%)	
Age in yrs., median (IQR)	65 (21)	66 (21)	39 (30)	45 (-)	67 (16)	44 (30)	63.5 (45)	74 (-)	*p* = 0.005 ^1–3†^ *p* = 0.002 ^2–3†^ *p* = 0.002 ^3–7†^
Gender—male, n (%)	83 (68.6)	43 (57.3)	21 (80.8)	1 (100)	23 (62.2)	4 (100)	1 (25)	1 (50)	*p* = 0.019 ^3–7^*
Risk group, n (%)									
Low	52 (43)	42 (56)	15 (57.7)	1 (100)	20 (54.1)	3 (75)	1 (25)	2 (100)	*p* > 0.05
Moderate	22 (18.2)	13 (17.3)	2 (7.7)	0	4 (10.8)	1 (25)	2 (50)	0	*p* > 0.05
High	47 (38.8)	20 (26.7)	9 (34.6)	0	13 (35.1)	0	1 (25)	0	*p* > 0.05
Outcome at 30 days—death, n (%)	Single valves IE—53 (23.8)				Double valves IE—14 (31.1)				*p* > 0.05
	28 (23.1)	19 (25)	5 (19.2)	1 (100)	10 (27)	3 (75)	1 (25)	2 (100)	*p* = 0.018 ^1–6^* *p* = 0.029 ^2–6^* *p* = 0.019 ^3–6^* *p* = 0.012 ^1–8^* *p* = 0.018 ^2–8^* *p* = 0.011 ^3–8^* *p* = 0.029 ^5–8^*
EF %, median (IQR)	60 (13)	60 (13)	61.5 (20)	0	64 (14)	56.5 (12)	65 (10)	61 (-)	*p* > 0.05
Type of valves									
Native IE	74 (61.2)	56 (74.7)	21 (80.8)	1 (100)	24 (64.9)	4 (100)	3 (75)	0	*p* > 0.05 *
Prosthetic IE	47 (38.8)	19 (25.3)	5 (19.2)	0	13 (35.1)	0	1 (25)	0	*p* > 0.05 *
Late prosthetic	42 (34.7)	18 (24.0)	4 (15.4)	0	11 (29.7)	0	1 (25)	0	*p* > 0.05 *
Early prosthetic	5 (4.1)	1 (1.3)	1 (3.8)	0	2 (5.4)	0	0	0	*p* > 0.05 *
Early surgery, n (%)	24 (19.8)	10 (13.3)	6 (23.1)	0	11 (29.7)	0	1 (25)	2 (100)	*p* = 0.036 ^2–5^* *p* = 0.001 ^2–8^* *p* = 0.006 ^1–8^* p = 0.001 ^2–8^ *p* = 0.020 ^3–8^* *p* = 0.040 ^5–8^*
Entry point									
Unknown	64 (52.9)	36 (48)	4 (15.4)	0	17 (46)	2 (50)	2 (50)	0	*p* = 0.001 ^1–3^* *p* = 0.003 ^2–3^* *p* = 0.011 ^3–5^*
Dental	15 (12.4)	8 (10.7)	0	0	7 (18.9)	0	0	0	*p* > 0.05 *
Skin	4 (3.3)	3 (4.0)	1 (3.8)	0	2 (5.4)	0	0	0	*p* > 0.05 *
Hemodialysis	4 (3.3)	6 (8.0)	0	0	1 (2.7)	1 (25)	1 (25)	0	*p* = 0.029 ^1–6^ *p* = 0.029 ^1–7^* *p* = 0.049 ^5–6^* *p* = 0.049 ^5–7^*
Urogenital	8 (6.6)	0	1 (3.8)	0	0	0	0	0	*p* > 0.05
Gastrointestinal	1 (0.8)	0	0	0	3 (8.1)	0	1 (25)	0	*p* = 0.013 ^1–5^* *p* = 0.000 ^1–7^*
Ear/nose/throat	1 (0.8)	3 (4.0)	0	0	0	0	0	0	*p* > 0.05
IV drug use	5 (4.1)	2 (2.7)	16 (61.6)	0	0	1 (25)	0	0	*p* < 0.0001 ^1–3^* *p* < 0.0001 ^2–3^* *p* = 0.024 ^2–6^*
Manipulation/ procedures	17 (14)	15 (20)	4 (15.4)	1 (100)	6 (16.2)	0	0	2 (100)	*p* = 0.001 ^1–8^* *p* = 0.007 ^2–8^* *p* = 0.002 ^3–8^* *p* = 0.004 ^5–8^*
Respirators	2 (1.7)	2 (2.7)	0	0	1 (2.7)	0	0	0	*p* > 0.05 *
Predisposing Heart Conditions									
Prosthetic valve	47 (38.8)	19 (25.3)	5 (19.3)	0	13 (35.1)	0	1 (25)	0	*p* > 0.05 *
Rheumatic heart disease	2 (1.7)	4 (5.3)	0	0	2 (5.4)	1 (25)	0	0	*p* = 0.003 ^1–6^*
Congenital heart disease	1 (0.8)	0	1 (3.8)	0	0	0	0	0	*p* > 0.05 *
Degenerative valve	10 (8.3)	3 (4)	1 (3.8)	0	1 (2.7)	0	0	0	*p* > 0.05 *
Bicuspid Aortic valve	9 (7.4)	0	0	0	1 (2.7)	0	0	0	*p* > 0.05 *
Mitral valve prolapse	0	5 (6.7)	0	0	0	0	1 (25)	0	*p* > 0.05 *
Without	52 (43)	44 (58.7)	19 (73.1)	1(100)	20 (54.1)	3 (75)	0	0	*p* > 0.05 *
Type of acquisition									
Community-acquired IE	87 (71.9)	52 (69.3)	7 (26.9)	0	26 (70.3)	1 (25)	2 (50)	0	*p* < 0.0001 ^1–3^* *p* = 0.0002 ^2–3^* *p* = 0.0007 ^3–5^* *p* = 0.043 ^1–6^*
Health-care-associated IE	29 (24)	22 (29.3)	4 (15.4)		11 (29.7)	1 (25)	2 (50)	2 (100)	*p* = 0.014 ^1–8^* *p* = 0.033 ^2–8^* *p* = 0.005 ^3–8^* *p* = 0.040 ^5–8^*
Intravenous-drug-use-related IE	5 (4.1)	1 (1.3)	15 (57.7)	0	0	2 (50)	0	0	*p* < 0.0001 ^1–3^*

* z test; ^†^ Mann–Whitney U test; AV—aortic valve; MV—mitral valve; TV—tricuspid valve; PV—pulmonic valve; AV + MV—bivalve aortic and mitral IE; AV + TV—bivalve aortic and tricuspid IE; MV + TV—bivalve mitral and tricuspid IE; CDRIE—intracardiac-device-related IE; IV drug use—intravenous drug use; EF—injection fraction.

**Table 2 microorganisms-12-00987-t002:** Comorbidity and complications.

Variables	AV	MV	TV	PV	AV + MV	AV+ TV	MV + TV	CDRIE	*p*-Value
	n = 121	n = 75	n = 26	n = 1	n = 37	n = 4	n = 4	n = 2	
	1	2	3	4	5	6	7	8	
	n (%)	n (%)	n (%)	n (%)	n (%)	n (%)	n (%)	n (%)	
Comorbidity									
CCI, median (IQR)	3 (3)	4 (4)	1 (5)	1(-)	4 (3)	2.5 (3)	3.5 (3)	3 (-)	*p* = 0.048 ^1–3†^ *p* = 0.011 ^2–3†^
AH	84 (69.4)	52 (69.3)	8 (30.8)	1(100)	21 (56.8)	1 (25)	3 (75)	1 (50)	*p* = 0.0002 ^1–3^* *p* = 0.001 ^2–3^* *p* = 0.041 ^3–5^*
CAD	37 (30.6)	17 (22.7)	2 (7.7)	0	6 (16.2)	0	1 (25)	1 (50)	*p* = 0.016 ^1–3^*
Heart surgery	48 (39.7)	25 (33.3)	9 (34.6)	0	12 (32.4)	0	1 (25)	0	*p* > 0.05 *
CHF	59 (48.8)	32 (42.7)	17 (65.4)	0	14 (37.8)	1 (25)	1 (25)	0	*p* > 0.05 *
Diabetes	22 (18.2)	17 (22.7)	3 (11.5)	0	8 (21.6)	1 (25)	0	0	*p* > 0.05 *
Atrial fibrillation	22 (18.2)	14 (18.7)	4 (15.4)	0	7 (18.9)	0	1 (25)	1 (50)	*p* > 0.05 *
Gastrointestinal	16 (13.2)	6 (8)	2 (7.7)	0	7 (18.9)	0	1 (25)	0	*p* > 0.05 *
Malignancy	11 (9.1)	10 (13.3)	2 (7.7)	0	7 (18.9)	0	0	0	*p* > 0.05 *
Systemic disease	1 (0.8)	3 (4.0)	0	0	0	0	0	0	*p* > 0.05 *
CKD	35 (28.9)	24 (32)	3 (11.5)	0	5 (13.5)	2 (50)	1 (25)	0	*p* > 0.05 *
Hemodialysis	5 (4.1)	6 (8.0)	0	0	1 (2.7)	1 (25)	1 (25)	0	*p* = 0.049 ^5–6^* *p* = 0.049 ^5–7^*
Chronic liver disease	7 (5.8)	1 (1.3)	5 (19.2)	0	0	0	0	0	*p* = 0.024 ^1–3^* *p* = 0.001 ^2–3^*
Past stroke	20 (16.5)	13 (17.3)	3 (11.5)	0	4 (10.8)	0	0	0	*p* > 0.05 *
Past IE	5 (4.1)	3 (4.0)	7 (26.9)	0	4 (10.8)	0	1 (25)	0	*p* = 0.0001 ^1–3^* *p* = 0.0008 ^2–3^*
Complications									
AHF	57 (47)	31 (41.3)	10 (38.5)	1 (100)	13 (35.1)	2 (50)	2 (50)	1 (50)	*p* > 0.05 *
Septic shock	11 (9.1)	5 (6.7)	2 (7.7)	0	2 (5.4)	3 (75)	0	0	*p* < 0.0001 ^1–6^* *p* < 0.0001 ^2–6^* *p* = 0.001 ^3–6^* *p* < 0.0002 ^5–6^*
Embolism	31 (20.7)	14 (18.7)	6 (15.4)	0	12 (32.4)	1 (25)	0	0	*p* > 0.05 *
Brain	15 (12.4)	7 (9.3)	0	0	6 (16.2)	1 (25)	0	0	*p* > 0.05 *
Lung	0	0	3 (11.5)	0	0	1 (25)	1 (25)	0	*p* > 0.05 *
Spleen	3 (2.5)	3 (4.0)	0	0	2 (5.4)	1 (25)	1 (25)	0	*p* = 0.012 ^1–6^* *p* = 0.012 ^1–7^*
Other	5 (4.1)	3 (4.0)	1 (3.8)	0	3 (8.1)	0	0	0	*p* > 0.05 *
Worsening kidney function	54 (44.6)	28 (37.3)	11 (42.3)	0	14 (37.8)	3 (75)	1 (25)	0	*p* > 0.05 *
Stroke	14 (11.6)	8 (10.7)	1 (3.8)	0	6 (16.2)	1 (25)	0	0	*p* > 0.05 *

* z test; ^†^ Mann–Whitney U test; AV—aortic valve; MV—mitral valve; TV—tricuspid valve; PV—pulmonic valve; AV + MV—bivalve aortic and mitral IE; AV+ TV—bivalve aortic and tricuspid IE; MV + TV—bivalve mitral and tricuspid IE; CDRIE—intracardiac-device-related IE; CCI—Charlson comorbidity index; AH—arterial hypertension; CAD—coronary arterial disease; CHF—chronic heart failure; CKD—chronic kidney disease; AHF—acute heart failure.

**Table 3 microorganisms-12-00987-t003:** Microbiological agents.

Microbiological Agent	AV	MV	TV	PV	AV + MV	AV+ TV	MV + TV	CDRIE	*p*-Value
	n = 121	n = 75	n = 26	n = 1	n = 37	n = 4	n = 4	n = 2	
	1	2	3	4	5	6	7	8	
	n (%)	n (%)	n (%)	n (%)	n (%)	n (%)	n (%)	n (%)	
Negative blood cultures	61 (50.4)	26 (34.7)	6 (23.1)	0	13 (35.1)	1 (25)	1 (25)	0	*p* = 0.011 ^1–3^*
*Staphylococci*	30 (24.8)	34 (45.3)	14 (53.9)	0	7 (18.9)	2 (50)	1 (25)	1 (50)	*p* = 0.003 ^1–2^* *p* = 0.003 ^1–3^* *p* = 0.006 ^2–5^* *p* = 0.004 ^3–5^*
*Staphylococcus aureus*	14 (11.6)	17 (22.6)	12 (46.2)	0	6 (16.2)	1 (25)	1 (25)	0	*p* < 0.0001 ^1–3^* *p* = 0.022 ^2–3^* *p* = 0.009 ^3–5^* *p* = 0.015 ^1–2^*
*Staphylococcus CoNS*	16 (13.2)	17 (22.7)	2 (7.7)	0	1 (2.7)	1 (25)	0	1 (50)	*p* = 0.007 ^2–5^* *p* = 0.003 ^5–8^*
*Streptococci*	8 (6.6)	6 (8.0)	0	0	7 (18.9)	0	0	0	*p* = 0.025 ^1–5^*
*Streptococcus viridans*	3 (2.5)	3 (4.0)	0	0	3 (8.1)	0	0	0	*p* > 0.05 *
*Streptococcus beta-hemolytic*	1 (0.8)	1 (1.3)	0	0	0	0	0	0	*p* > 0.05 *
*Streptococcus alfa hemolyt*	2 (1.7)	1 (1.3)	0	0	3 (8.1)	0	0	0	*p* > 0.05 *
*Streptococci—other*	2 (1.7)	1 (1.3)	0	0	1 (12.7)	0	0	0	*p* = 0.004 ^1–5^* *p* = 0.010 ^2–5^*
*Enterococci*	9 (7.4)	5 (6.7)	2 (7.7)	1 (100)	4 (10.8)	1 (25)	2 (50)	1 (50)	*p* = 0.003 ^1–7^* *p* = 0.003 ^2–7^* *p* = 0.021 ^3–7^* *p* = 0.035 ^5–7^* *p* = 0.028 ^1–8^* *p* = 0.025 ^2–8^*
*Enterococcus species*	0	0	0	0	1 (2.7)	0	0	0	
*Enterococcus faecalis*	9 (7.4)	5 (6.7)	2 (7.7)	1 (100)	2 (5.4)	1 (25)	2 (50)	1 (100)	*p* = 0.003 ^1–7^* *p* = 0.003 ^2–7^* *p* = 0.021 ^3–7^* *p* = 0.004 ^5–7^* *p* < 0.0001 ^1–8^* *p* < 0.0001 ^2–8^* *p* = 0.0003 ^3–8^* *p* < 0.0001 ^5–8^*
*Enterococcus durans*	0	0	0	0	1 (2.7)	0	0	0	
*Gram-negative (non-HASEK)*	12 (9.9)	3 (4.0)	3 (11.5)	0	1 (2.7)	0	0	0	*p* > 0.05 *
*Pseudomonas aeruginosa*	0	1 (1.3)	0	0	1 (2.7)	0	0	0	*p* > 0.05 *
*Escherichia coli*	6 (5.0)	1 (1.3)	2 (7.7)	0	0	0	0	0	*p* > 0.05 *
*Enterobacter cloacae*	1 (0.8)	0	0	0	0	0	0	0	-
*Klebsiella pneumoniae*	2 (1.7)	1 (1.3)	0	0	0	0	0	0	*p* > 0.05 *
*Serratia marcescens*	3 (2.5)	0	1 (3.8)	0	0	0	0	0	*p* > 0.05 *
*Others*	1 (0.8)	0	0	0	3 (8.1)	0	0	0	*p* = 0.013 ^1–5^*
*Candida spp.*	0	0	0	0	3 (8.1)	0	0	0	-
*Erysipelothix rhusiopathiae*	0	1 (1.3)	0	0	0	0	0	0	-
*Brevibacterium casei*	1 (0.8)	0	0	0	0	0	0	0	-
*Missing*	0	0	1 (3.8)	0	2 (5.4)	0	0	0	-

* z-test.

## Data Availability

The raw data supporting the conclusions of this article will be made available by the authors on request.

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
