# Peer review of "Infective Endocarditis—Characteristics and Prognosis According to the Affected Valves"

_microorganisms, 2024, doi:10.3390/microorganisms12050987_

Round 1

Reviewer 1 Report

Comments and Suggestions for Authors

1. You claim that there are no studies comparing all types of valve involvement in IE. Here are some, just some, there are more in the literature, you just have to focus beyond the abstracts: Netzer et al-Heart 2002, Pereira Nunes et al-IJID 2018, Moreira et al-Rev Port Cardiol 2018, Leroy et al-Ann Intensive Care 2015. There are more. 

2. How many cases were ultrasonographically negative? I deduce that it was 1, since your distribution in valves adds to 269 instead of 270? Or should it be 268 since you have 2/270 cases of CDRIE? What kind of devices were implicated in these two cases?

3. Proclaiming that Canada, India and Iran have a lot of iv drug abusers, based on isolated studies from these countries that may focus on a limited number of patients or geographic region is improper, to say the least. If I was a Canadian or an Indian or an Iranian I would be insulted. Do you have data that show that iv drug addiction is widespread, as you state, in these countries?

Author Response

Dear Reviewer,

    Thank you very much for taking the time to review this manuscript. Thank you for the constructive review, the referee’s comments have improved our manuscript. Please find the detailed responses below and the corresponding revisions/corrections highlighted changes in the re-submitted files. 

 Questions for General

Reviewer’s Evaluation

Response and Revisions

Does the introduction provide

sufficient background and

include all relevant references?

Are all the cited references

relevant to the research?

Is the research design

appropriate?

Are the methods adequately

described?

Are the results clearly presented?

Are the conclusions supported by

the results?

Must be improved

Must be improved

Can be improved

Can be improved

Must be improved

Must be improved

Has been improved

Has been improved

Has been improved

Has been improved

Has been improved

Has been improved

Comment

Respons

Location in revised MS

1.     You claim that there are no studies   comparing all types of valve involvement in IE. Here are some, just some, there are more in the literature, you just have to focus beyond the abstracts: Netzer et al-Heart 2002, Pereira Nunes et al-IJID 2018, Moreira et al-Rev Port Cardiol 2018, Leroy et al-Ann Intensive Care 2015. There are more

1.We agree and have rewritten and expanded this section to clarify our statements based on previous studies.

Introduction

2.     How many cases were ultrasonographically negative? I deduce that it was 1, since your distribution in valves adds to 269 instead of 270? Or should it be 268 since you have 2/270 cases of CDRIE? What kind of devices were implicated in these two cases?

2.Thank you for paying attention. This is a technical error - the total single valve cases  is 223 (82.6%), double valves are 45 (16.66%) and 2 (0.74%) CDRIE cases. The total valves distribution is 268. CDRIE are both permanent VVI pacemaker.

Table. 1

3.     Proclaiming that Canada, India and  

Iran have a lot of iv drug abusers, based on isolated studies from these countries that may focus on a limited number of patients or geographic region is improper, to say the least. If I was a Canadian or an Indian or an Iranian I would be insulted. Do you have data that show that iv drug addiction is widespread, as you state, in these countries?

    3. We accept your remark    

    as valid.

    We have rewritten the      

    comment about tricuspid   

    valve involvement.

Manuscript  Line  226 - 232

Reviewer 2 Report

Comments and Suggestions for Authors

Thank you for inviting me to review this manuscript. It is interesting and informative. I have some comments that could be of use:

1.     Line 27: It is not clear what the numbers in the parentheses are

2.     Line 29: All gender and species names should be in italics throughout the manuscript

3.     Line 48: A dot is missing

4.     The aim of the present study should be more clearly stated at the end of the introduction section

5.     The authors mention that there are no studies comparing multivalvular with no multivalvular endocarditis. That is not correct. There are studies such as doi: 10.3390/jcm11164736 or 10.1016/j.cjca.2020.03.046. These studies should be mentioned in the introduction section

6.     This manuscript has too many sections. The structure should be: introduction, methods, results, and discussion

7.     The authors say that this is a cross-sectional study. Isn’t this a retrospective study?

8.     Table 1: the table should have an explanation for all abbreviations used in the footnote

9.     Table 3: The percentages of CoNS and aureus do not always add up to the percentage mentioned in all staphylococci. Especially in AV. Why is that?

10.  This study lacks a limitations subsection. It has several drawbacks, such as its obvious retrospective nature, the fact that it has data from only one center, limiting the generalization of the results, etc. This should be mentioned at the end of the discussion section before the conclusions

Comments on the Quality of English Language

Minor

Author Response

Dear Reviewer,

  Thank you very much for taking the time to review this manuscript. Thank you for the constructive review, the referee’s comments have improved our manuscript. Please find the detailed responses below and the corresponding revisions/corrections highlighted changes in the re-submitted files. 

 Questions for General

 Reviewer’s Evaluation

Response and Revisions

Does the introduction provide

sufficient background and

include all relevant references?

Are all the cited references

relevant to the research?

Is the research design

appropriate?

Are the methods adequately

described?

Are the results clearly presented?

Are the conclusions supported by

the results?

Must be improved

Can be improved

Can be improved

Can be improved

Must be improved

Can be improved

Has been improved

Has been improved

Has been improved

Has been improved

Has been improved

Has been improved

Comments

Respons

Location in revised MS

1. Line 27: It is not clear what the numbers in the parentheses are

The Charlson comorbidity index is present as a median (IQR).

We changed  than with vs

Line 27

2. Line 29: All gender and species names should be in italics throughout the manuscript

Yes, Thank you!

I made them in italics.

Line 24,29,30,31,34

3. Line 48: A dot is missing

We added it.

Line 48

4. The aim of the present study should be more clearly stated at the end of the introduction section

Thank you! We made the change throughout the manuscript.

Introduction

5. The authors mention that there are no studies comparing multivalvular with no multivalvular endocarditis. That is not correct. There are studies such as doi: 10.3390/jcm11164736 or 10.1016/j.cjca.2020.03.046. These studies should be mentioned in the introduction section

We appreciate your comment. We have rewritten the introduction and included these studies.

Introduction

6. This manuscript has too many sections. The structure should be introduction, methods, results, and discussion

We agree with this comment and  have restructured the manuscript.

7. The authors say that this is a cross-sectional study. Isn’t this a retrospective study?

You have the right; it is our mistake. We deleted cross-sectional.

Line 18, 52

8. Table 1: the table should have an explanation for all abbreviations used in the footnote

Yes, thank you!

It is our laps.

We have written them.

Table 1, footnote

Line 98-101

9. Table 3: The percentages of CoNS and aureus do not always add up to the percentage mentioned in all staphylococci. Especially in AV. Why is that?

Thank you for paying attention. This is a technical error – the number of patients is correct, but the percentage of Staphylococcus aureus – 9.9% is not correct . The right percent is 11.6%.  We have changed it.

Table 3.

10. This study lacks a limitations subsection. It has several drawbacks, such as its obvious retrospective nature, the fact that it has data from only one center, limiting the generalization of the results, etc. This should be mentioned at the end of the discussion section before the conclusions

We agree, thank you!

We added this section.

Section Limitations

Round 2

Reviewer 1 Report

Comments and Suggestions for Authors

Satisfied by the responses

Reviewer 2 Report

Comments and Suggestions for Authors

The manuscript has been improved during the revisions

Comments on the Quality of English Language

Minor